# Is stock market development sensitive to macroeconomic indicators? A fresh evidence using ARDL bounds testing approach

Abdullah Bin Omar[1]*, Anis Ali[2], Salma Mouneer[3], Robina Kouser[4], Mamdouh Abdulaziz Saleh Al-Faryan[5,6]

1 Department of Business Administration, National College of Business Administration & Economics (NCBA&E), Lahore, Pakistan, 2 Department of Management, College of Business administration, Prince Sattam Bin Abdulaziz University, Al Kharj, Saudi Arabia, 3 Department of Economics, Women University, Multan, Pakistan, 4 Department of Economics, University of Sahiwal, Sahiwal, Pakistan, 5 Faculty of Business and Law, School of Accounting, Economics and Finance, University of Portsmouth, Portsmouth, United Kingdom, 6 Consultant in Economics and Finance, Riyadh, Saudi Arabia

* abdullah@ncbaemultan.edu.pk

**Data Availability Statement:** All relevant data is already available in the paper.

**Funding:** The authors received no specific funding for this work.

## Abstract

Over the past decades, emerging stock markets have started to significantly contribute to economic growth through mobilizing long-term capital by pooling funds, facilitating savings and investments into profitable projects and improving corporate governance structure. A plethora of empirical studies is devoted to investigate the determinants of different capital markets but due to highly controversial and inconclusive findings about macroeconomic determinants, this study contributes to the body of existing literature by empirically investigating the macroeconomic forces that drive the stock market development of Pakistan from 1980 to 2019. By applying Ng-Perron and Zivot-Andrews unit root tests (to determine the integrating orders of variables) and Autoregressive Distributed Lag (ARDL) bounds testing approach, our results confirm cointegration among variables and exhibit the significant positive impact of economic growth and banking sector development on stock market development and negative affect of inflation, foreign direct investment and trade openness on it in long run. At the same time, the short run results show a significant relationship of economic growth, inflation and foreign direct investment with stock market development. Our study has some important policy implications.

## 1. Introduction

In any economy, the capital market plays a significant role as an integral part of the financial system by channelizing the investments of public and private entities into profitable projects, mobilizing savings by efficiently allocating the financial resources and assisting the redistribution of those financial resources among individuals, corporations and governments. During the last decade, the vital role of capital markets has become dominant and nations' growth is now heavily dependent on stock markets [1]. An established financial and capital market is

**Competing interests:** The authors have declared that no competing interests exist.

likely to affect economic growth by promoting the development of capital markets and hedging instruments, fostering the savings level, improving resource allocation and mitigating transaction and information costs [2]. Existing theoretical economic and financial literature expose that the deep-rooted capital market promotes international risk-sharing, improves governance structure, mitigates the cost of mobilizing resources, and provides market liquidity [3–6]. Furthermore, capital markets also provide a platform for trading listed securities and a significant source of mobilizing liquidity and capital. That's why well-established financial markets provide a sufficient justification for why sound and developed economies remain developed.

This study aims to provide empirical evidence on macroeconomic forces that influence the performance of the stock market of Pakistan in terms of its capitalization in the presence of unknown structural breaks during 1980–2019 using time-series data. In Pakistan, the Securities & Exchange Commission of Pakistan (SECP) serves as the apex regulator of the Pakistani equity market to protect investors' interests and ensure smooth functioning by regulating and overseeing the performance of capital markets and their intermediaries. The capital market of Pakistan is comprised of the PSX, the Central Depository Company (CDC), the National Clearing Company of Pakistan Limited (NCCPL), and the Pakistan Mercantile Exchange Limited (PMEX) [7].

In literature, the Pakistan stock exchange (PSX) has been subject to great interest due to several reasons, such as its remarkable growth and its outstanding performance in the Asian and global capital markets. Notably, the historic day in the history of Pakistan is January 11, 2016, when Pakistan entered a new era with the formal integration of three stock exchanges, namely Karachi Stock Exchange, Islamabad Stock Exchange, and Lahore Stock Exchange into PSX [8]. This greater reform provided a fully integrated national equity platform to all market players (including foreign institutional investors) in the Pakistani capital market with a single deep liquidity pool and enhanced its operational efficiency. Secondly, according to Bloomberg, in 2016, the Pakistan Stock Exchange secured its fifth position as the best performing capital market in the world and the best capital market in Asia [7]. Thirdly, in Frontier Markets classified by Morgan Stanley Capital International (MSCI), Pakistan also had the top favorable return of 46% as compared to an average return of 20% over the past ten years [7] which resultantly upgraded the Pakistan economy from the status of frontier markets to emerging markets by the MSCI on June 14, 2016 [9]. This became possible mainly due to exceptional performance demonstrated by the KSE-100 index, improved reforms introduced by SECP over the years, friendly investment policies of a government, and improved macroeconomic indicators.

Furthermore, in 2020, several adverse economic conditions push the Pakistan economy into different challenges, such as the oil price crash, austerity measures, and high inflation. Particularly, the outbreak of the Covid-19 pandemic jolted the global equity market and plummeted the PSX. In response, several steps undertook by the SECP to absorb the existing economic shocks, stabilize the capital market and rebound the economy. For example, during the normal operations of PSX, securities brokers are allowed to activate the Disaster Recovery Terminals for uninterrupted trading. Similarly, PSX increased the duration of index-based market halts from 45 minutes to 60 minutes. Furthermore, the market cost was also reduced by SECP along with rationalizing the fee structure of the PSX, CDC and NCCPL to maximum facilitate the market players [10]. Due to these notable rebounded reforms, PSX secured 4[th] position as a best-performer market across the global capital market and got the title of 'best Asian stock market' in 2020 [11].

During the second wave of Covid-19, PSX outperformed and retained its dominant position among Asian capital markets in 2020–21. For example, an all-time high daily trading volume of 2.21 billion shares was recorded in a single session on May 27, 2021. Furthermore, the

KSE-100 index increased by 13,006 points from 34,889 to 47,896 points in 2021. At the end of the said period, an increase of 26.6 percent was witnessed in the total market capitalization of the PSX [10]. Although the third wave of Covid-19 pulled the KSE-100 index down in the first quarter of 2021, however, the SECP reforms and proactive government policies are holding up the capital market to withstand the pressure.

Given the said facts and the significant role of PSX played among Asian and global capital markets, a better understanding and greater knowledge about macroeconomic forces affecting its development warrant us to undertake this study. Therefore, to draw an estimation of macroeconomic factors for stock market development (hereinafter SMD), a case of PSX is selected and analyzed in more depth by applying the ARDL bounds testing approach. Our findings confirm the long run relationship among variables. Results reveal that economic growth and banking sector development improves the stock market performance, whereas it is negatively pronounced by inflation, foreign direct investment (FDI) and trade openness. Moreover, the result of the error correction term shows the hasty annual convergence towards equilibrium of a variable in case of facing economic shock in the short run.

## 2. Literature review

The influencing role of capital markets has prompted empirical research to investigate factors that drive the stock market. Resultantly, literature spawned several empirical studies examining the effect of institutional and macroeconomic determinants on SMD. As a whole, relevant past studies found that major macroeconomic factors such as economic growth, money supply, banking sector development, trade openness, stock market liquidity, foreign portfolio investment, inflation, domestic investment, private capital flows and saving rate significantly influence the SMD in both developing and developed economies [1,12–20]. This section provides a review of theoretical and empirical studies about macroeconomic determinants and SMD.

A consensus exists in theoretical literature that real income level is positively associated with financial sector development which, in result, trigger the capital market development [3,6,21]. When an economy grows, more people started to involve in financial activities and gain benefits by trading in financial products and services [4,22–24]. Similarly, the theoretical arguments about the relationship between banking sector and stock market development are largely inclusive as some argue that both are complement to each other while some argue that they are substitute to each other. Regarding the substitutability between two, some studies argued that banks outperform stock markets in execution of financial functions to the economy such as intertemporal risk sharing and information acquisition about firms [25–31]. While, on the other hand, Levine [32] argues that banking sector and stock market jointly provide financial services by playing their key role in boosting market liquidity and facilitating risk management services which ultimately foster the economic growth.

The theoretical literature on inflation signifies that smaller and less liquid equity markets are a cause of higher inflation rates in an economy and there is a nonlinear relationship exists between stock market development and inflation rate [33–37]. Likewise, theories on trade openness argued that it benefits the financial development through the supply side of capital market [38,39] and foster the stock market by enhancing the demand for financial products and services [40,41]. For the FDI, two opposite views in theoretical literature exists. According to Hausmann and Fernandez-Arias [42], Fernández-Arias and Hausmann [43], FDI negatively correlates with stock market development in developing economies that are institutionally weaker and risker; hence it works as an substitute for equity and debt financing in underdeveloped economies. On the other hand, Claessens, Klingebiel [44] claim that FDI strengthen the

financial systems of an economy having a sound institutional and financial system by improving the firms' participation in stock market. The cash flows of domestic stock markets can be increased with the help of FDI through buying and selling the existing securities by foreign investors.

A plethora of empirical studies in financial market literature are devoted to investigate the determinants of different capital markets. Thaddeus, Ngong [45] conducted their study on sub-Saharan Africa from 1990 to 2020 and investigated the short and long run causal relationship between economic growth and SMD. Their results reveal that stock market capitalization positively and significantly influences economic growth in the long run, while short run results are insignificant. However, their Granger causality results are inconclusive on whether the SMD causes economic growth or vice versa. A recent study on Ghana undertook by Asravor and Fonu [12] examines the long and short run relationship between macroeconomic variables and stock market returns and development from 1992 to 2017. Their findings unveil that the SMD is negatively affected by human capital, inflation rate and money supply and positively affected by the interest rate and FDI.

Using the Vector Error Correction Model (VECM), Olokoyo, Ibhagui [46] investigated the long run impact of macroeconomic variables on SMD in Nigeria from 1981 to 2018. Results show the positive effect of foreign capital flows, GDP, and exchange rate and the negative effect of trade openness, inflation and interest rate on SMD. By applying the Feasible Generalized Least Squares estimator on a panel data of the Sub-Saharan Africa during 2000 to 2015, Aluko and Kolapo [47] investigated the impact of macroeconomic factors on SMD and found that investment, savings, financial development, macroeconomic instability, financial openness, trade openness and income significantly influence the SMD. Ho [13] used the ARDL bounds testing approach and investigated the macroeconomic determinants for the South African stock market during 1975 to 2015 and found that real interest rate, inflation rate, and trade openness inhibit the SMD while economic growth and banking sector development promote the SMD. In her related study, Ho [48] examined the macroeconomic determinants of the Malaysian stock market from 1981 to 2015 and found that banking sector development has a significant negative impact on SMD in the long run and positive impact in the short run while trade openness and economic performance have a positive relationship in the long run with SMD.

Another study on the examination of macroeconomic indicators for the Turkish stock exchange conducted by Bayar [16] for the period of 2005–2015 using the ARDL cointegration approach found that the inflation decreases stock market capitalization in the long run while stock market liquidity and economic growth promote it. A related study conducted by Flannery and Protopapadakis [49] examined the impact of macroeconomic forces on the US capital market during 1980 to 1996 and found that inflation, money supply, housing, employment, balance of trade, producer price index significantly influenced the US stock market. The investigation of macroeconomic factors and their role in developing the stock market in selected European countries is explored by Şükrüoğlu and Nalin [17] using dynamic panel data for the period of 1995 to 2011. Results revealed that saving rate, liquidity ratio and income positively affect while inflation and monetization ratio negatively affect the SMD. In a relevant study, Owusu and Odhiambo [50] examined the relationship between economic growth and SMD in Ghana using the ARDL bounds testing approach. Their key finding is that capital account liberalization and SMD have no positive influence on economic growth in the long run.

The study of Yartey [18] explored the macroeconomic and institutional determinants of SMD using panel data of 42 developing countries over the period of 1990 to 2004 and found that stock market liquidity, private capital flows, banking sector development, gross domestic investment and income level are the important determinants of SMD in emerging economies.

Similar evidence on the influence of macroeconomic and institutional determinants of SMD is provided by Cherif and Gazdar [51] using data from 14 MENA countries during 1990 to 2007 and unveil the interest rate, capital market liquidity, saving rate, income level as the significant determinants of SMD.

Insofar the Pakistan economy is concerned, some studies attempted to examine macroeconomic determinants for the Pakistan stock exchange. Shahbaz, Rehman [15], for example, explored the macroeconomic variables influencing the SMD of Pakistan during 1974 to 2010 by applying ARDL bounds testing and the VECM approach. They found that trade openness inhibits the SMD while investment, financial development, inflation and economic growth foster it. A similar study was conducted by Shahbaz, Hooi Lean [52] to investigate the relationship between FDI and SMD in Pakistan and their findings support the complementary role of FDI in the SMD in Pakistan. In a similar fashion, Abdul Malik and Amjad [53] investigated the effect of FDI on SMD in Pakistan during 1985 to 2011 and found identical results. Raza, Jawaid [54] inspected the impact of economic growth and foreign capital inflows on SMD in Pakistan over the period of 1976 to 2011 and found their significant effect in both the long and short run on SMD.

The majority of the studies mentioned above focuses on developed economies, while developing countries are under-addressed. Similarly, empirical results explained above show that the findings of the studies are largely inconclusive about the macroeconomic determinants regarding the significance and influence on SMD. In the context of Pakistan, to the best of the authors' knowledge, the only time-series study carried out by Shahbaz, Rehman [15] examines the macroeconomic determinants of Pakistan's stock market. Given that few time-series studies are focusing on the macroeconomic determinants of the Pakistani capital market despite its remarkable growth, outstanding performance, and mega reforms in the last decade among Asian and global capital markets. This significant gap calls for a comprehensive and detailed investigation of macroeconomic forces that drive the SMD in Pakistan and the current study fills this gap.

## 3. Material and methods

### 3.1 Variables' description and measurement

• **Stock market development.** There is no consensus in the literature on a comprehensive definition of the phrase 'Stock Market Development' though it has been frequently used in past studies. In the empirical literature, it has been widely used in terms of size and liquidity of a stock market [55] and also in terms of volatility and level of international integration [56]. To remove complications, researchers started to define stock market development as total market capitalization scaled by gross domestic product (GDP) (also recognized as the Buffet Indicator) since this proxy is considered a good measurement of SMD as it is positively related to diversifying risk, provides the valuation of any stock market at any given moment and closely associated with capital mobilization [1,13,48,57,58]. To be consistent with these studies, we also used the market capitalization ratio as a proxy for SMD measured as the total market value of all stocks traded on the Pakistan Stock Exchange of listed domestic companies (% of GDP). Several studies, such as Demirgüç-Kunt and Levine [59], Aluko and Kolapo [47] and Levine and Zervos [60] argued that SMD is influenced by different underlying factors like the degree of international integration, a total number of listed firms, the liquidity, value of equity trading, total market capitalization including government securities, volatility, turnover ratio, and concentration and all of these factors are significantly correlated.

• **Economic growth.** In literature, there is a general consensus on the favorable effect of real income of an economy on financial market development which also incorporates the

development of stock markets [61–63]. Economic activities expand and reduced in a period of economic prosperity and recession, respectively. Theoretical studies have confirmed that rapid growth in the stock market system appears when an economy develops [64]. When an economy grows and is stable, the level of income and savings boost up. In turn, the stock market channelizes these savings into an investment which ultimately enhances the stock market capitalization. Hence, the synthesis of theoretical studies suggests the positive relationship between stock market development and economic growth, therefore we can also expect a positive association between the two. In this study, economic growth is proxied by GDP per capita.

• **Inflation.** Theoretical studies argue that high inflation rates are associated with small and less liquid equity markets [33–37]. Boyd, Levine [35] argue that the real rate of return on financial assets, particularly on money, is deteriorated by the increase in inflation rate which consequently reduces the lending incentive of agents. As a result, the credit opportunity is adversely affected and potential borrowers prefer to stay away from the credit-seeking pool. The high inflation rate reduced the savings level in the economy hence refrain the investment from moving towards the stock market, and stock market capitalization became reduced. So, we can expect the inverse relationship between inflation and SMD. In current study, inflation is proxied by the wholesale price index (2010 = 100).

• **Financial development.** Empirical literature provides inconclusive evidence about the relationship between the financial sector and stock market performance whether both are substitutes or complement each other. Several theoretical studies prove the substitutability between the two and show that, regarding the provision of financial services, the banking system performs better than capital markets [25–31]. In contrast, Levine [32] asserts the complementary relationship between financial sector development and stock market activities and argues that development in the banking sector may foster the stock market by boosting market liquidity and providing alternate means of financing investment. Apart from the substitute or complementary relationship between the two, aforesaid competing arguments and several empirical shreds of evidence connote that the established banking system in an economy fosters the development of stock market; therefore, we can expect a positive relationship between them. To measure financial development, we used a proxy of banking sector development measured as the domestic credit to the private sector (% of GDP).

• **Foreign direct investment.** In theoretical economic literature, two opposing views about FDI prevail. Some argue that FDI and SMD are substitute for each other [42,43]; because when FDI flows into an economy that is already financially weaker and underdeveloped, FDI becomes an alternate for developing financial markets about both equity and debt financing. Hence, a negative relationship can be expected between FDI and SMD. In contrast, some studies argue that they both complement to each other [44] as FDI flows foster the financial system of an economy that is already developed with sound and strong institutional infrastructure. As Pakistan is a developing economy with a growing institutional and financial system, we can expect a negative relationship between FDI and SMD. We measure the FDI as net inflows (% of GDP).

• **Trade openness.** From a theoretical perspective, trade openness strengthens the capital market from the "supply-side" [38,39,65] as well as the "demand-side" by boosting the demand for financial product and services [40,41]. On the supply-side, trade liberalization mitigates the influence of pressure groups that hinder financial market development. Therefore, when economies liberalize their trade sector policies, they experience developments in their capital markets. On the demand-side, trade openness increases volatilities in income due to an increase in price elasticities which, in turn, raises the demand for insurance related products, hence, fostering the SMD. Similarly, exposure to global trade competition enhanced due to trade openness which ultimately generates the demand for more financial products and services to

diversify different associated exposures and bring the boost in stock market capitalization. Therefore, we expect that the SMD is positively pronounced by trade openness and it is measured by imports plus exports as a % of GDP.

## 3.2 ARDL bounds testing approach

To examine the long run relationship between variables, we used the ARDL bounds testing approach to cointegration developed by Pesaran, Shin [66]. This approach is preferred over other traditional cointegration approaches (such as Engle and Granger [67] and Johansen and Juselius [68]) from several aspects. Traditional cointegration approaches require that all variables must be integrated of the same order. In contrast, the ARDL bounds testing method does not restrict the same order of integration and should be used even when variables are integrated of different orders like *I(0)*, *I(1)* or both [69]. Furthermore, while other traditional cointegration tests are sensitive to sample size, the ARDL bounds approach can be easily used even when the sample size is small and provides better results [13,70]. Linear specification of ARDL bounds testing model used for empirical investigation is given below:

$$
\begin{aligned}
\Delta \ln MC_t = \alpha_0 &+ \sum_{i=1}^{p}\alpha_{1i}\Delta \ln MC_{t-i} + \sum_{i=0}^{q}\alpha_{2i}\Delta \ln GDP_{t-i} + \sum_{i=0}^{q}\alpha_{3i}\Delta \ln INF_{t-i} \\
&+ \sum_{i=0}^{q}\alpha_{4i}\Delta \ln BNK_{t-i} + \sum_{i=0}^{q}\alpha_{5i}\Delta \ln FDI_{t-i} + \sum_{i=0}^{q}\alpha_{6i}\Delta \ln TRO_{t-i} \\
&+ \beta_{MC}\ln MC_{t-1} + \beta_{GDP}\ln GDP_{t-1} + \beta_{INF}\ln INF_{t-1} + \beta_{BNK}\ln BNK_{t-1} \\
&+ \beta_{FDI}\ln FDI_{t-1} + \beta_{TRO}\ln TRO_{t-1} + \gamma DUM_t + \mu_t
\end{aligned} \tag{1}
$$

Where, the notations, namely, *MC*, *GDP*, *INF*, *BNK*, *FDI* and *TRO* are used as a proxy for stock market development, economic growth, inflation, banking sector development, foreign direct investment and trade openness respectively. In addition, our dataset is likely to be affected by structural breaks during the sample period because of several factors. For example, an economic shock during the Global Financial Crises from mid-2007 to early-2009 may exist in our dataset which needs to be account for because investors carry different perceptions about stock market during and after the financial turbulences [71]. Moreover, during any financial crises, the investors are inclined to pull their investment out of the stock market because of economic uncertainty which resultantly upsurge the capital outflows and deteriorate the stock market [72]. Therefore, a dummy variable (*DUM*) is added into a model to capture the structural breaks which takes a value of 0 when there is no break and 1 when there is a break. $\alpha$, $\beta$, and $\mu_t$ are short run coefficients, long run coefficients, and white-noise error term respectively; $\Delta$ is the first differenced operator; *t* denote time period while *p* and *q* refer to the maximum number of lags used for dependent and exogenous variables respectively. Schwarz criterion (SC) is used to determine the maximum number of lags in the model. The selection of these macroeconomic factors is informed by theoretical as well as empirical studies discussed in the previous section.

To confirm whether or not cointegration exists among variables, the joint significance of long run coefficients, $\beta_{MC}$, $\beta_{GDP}$, $\beta_{INF}$, $\beta_{BNK}$, $\beta_{FDI}$, $\beta_{TRO}$, is examined by testing the null hypothesis of no cointegration relationship:

$$H_0: \ \beta_{MC} = \beta_{GDP} = \beta_{INF} = \beta_{BNK} = \beta_{FDI} = \beta_{TRO} = 0$$

Pesaran, Shin [66] tabulated two sets of critical values namely lower bound and upper bound. Lower and upper bound critical values are based on the assumption that variables in a

model are *I(0)* and *I(1)*, respectively. The results of calculated *F*-statistics need to be compared with upper and lower critical bounds. The cointegration is confirmed if the calculated *F*-statistics is greater than the upper critical bound; hence null hypothesis is rejected. Likewise, if the lower critical bound is greater than the calculated *F*-statistics, null hypothesis of no-cointegration cannot be rejected. Finally, if calculated *F*-statistics falls between the lower and upper critical bounds, the decision about cointegration remain inconclusive.

If cointegration among variables is confirmed, we'll further move towards determining the short-run relationships of the variables by adding an error-correction term (*ECT*) in a model stated below:

$$
\Delta \ln MC_t = \alpha_0 + \sum_{i=1}^{p} \alpha_{1i} \Delta \ln MC_{t-i} + \sum_{i=0}^{q} \alpha_{2i} \Delta \ln GDP_{t-i} + \sum_{i=0}^{q} \alpha_{3i} \Delta \ln INF_{t-i} + \sum_{i=0}^{q} \alpha_{4i} \Delta \ln BNK_{t-i} +
$$
$$
\sum_{i=0}^{q} \alpha_{5i} \Delta \ln FDI_{t-i} + \sum_{i=0}^{q} \alpha_{6i} \Delta \ln TRO_{t-i} + \beta_{ECT} ECT_{t-1} + \mu_t
$$

(2)

Where $\beta_{ECT}$ is the coefficient of *ECT*. If $\beta_{ECT}$ is negative and significant, then its value indicates the tendency of adjustment of variables towards equilibrium level after facing the shock in short-run.

Annual time-series data covers the period from 1980 to 2019 to examine the role of key macroeconomic factors (i.e., economic growth, inflation, financial development, FDI and trade openness) in explaining SMD. The selected time period is solely based on data availability. Data on SMD is obtained from the Datastream whereas the world development indicators (WDI) (compiled by the World Bank) is used to collect data for all explanatory variables (www.worldbank.org). To reduce the data sharpness, we transformed all variables' series into natural logarithm form which is likely to generate reliable results by reducing the variance in series, mitigating the effect of outliers and enabling policymakers to understand the influence of macroeconomic factors on SMD.

## 4. Results & discussion

### 4.1 Descriptive statistics

Descriptive statistics of SMD and macroeconomic factors effecting its performance are illustrated in Table 1. The high standard deviation in inflation rate (*INF*) is 0.99 indicating the high volatilities in wholesale price index throughout the sample period which has a likely negative impact on stock market. With the minimum value of -2.27, a maximum value of 1.29 and the mean value of *FDI* is -0.38 implying that foreign investment squeezes the domestic businesses from the market which has possibly the negative impact on stock market. With the average of 6.41, the growth rate in GDP of a country is low which reveals the slow economic growth in economic activities of Pakistan. Furthermore, minimum value 5.71 and maximum value 11.30 of *GDP* shows the volatility in economic development of Pakistan which is expected to have a negative effect on Pakistan stock market. Pair-wise correlations are also given in Table 1. It is shown that, consistent with economic theories, economic growth, banking sector development, FDI and trade openness are positively associated, whereas inflation is negatively associated with SMD. Finally, there is a negative association of banking sector development and trade openness with economic growth, and inflation.

**Table 1. Descriptive statistics and correlation matrix of the variables.**

| | *MC* | *GDP* | *INF* | *BNK* | *FDI* | *TRO* |
|---|---|---|---|---|---|---|
| **Mean** | 4.773500 | 6.418000 | 3.619147 | 3.115278 | -0.380062 | 3.478147 |
| **Median** | 4.835000 | 6.245000 | 3.651457 | 3.182144 | -0.436537 | 3.495451 |
| **Maximum** | 5.770000 | 11.300000 | 5.193026 | 3.394041 | 1.299735 | 3.650640 |
| **Minimum** | 3.950000 | 5.710000 | 2.027819 | 2.733463 | -2.276267 | 3.231051 |
| **Std. Dev.** | 0.450006 | 0.520637 | 0.992773 | 0.183497 | 0.738730 | 0.110976 |
| **Skewness** | 0.001043 | 0.387328 | 0.014113 | -0.674597 | 0.072120 | -0.708298 |
| **Kurtosis** | 2.811760 | 1.651665 | 1.734804 | 2.317418 | 3.579106 | 2.750480 |
| **Sum Sq. Dev.** | 7.897710 | 10.57144 | 38.43832 | 1.313168 | 21.28314 | 0.480312 |
| **Observations** | 40 | 40 | 40 | 40 | 40 | 40 |
| *Correlation Matrix* | | | | | | |
| *MC* | 1.000000 | | | | | |
| *GDP* | 0.356164 | 1.000000 | | | | |
| *INF* | -0.467858 | 0.971716 | 1.000000 | | | |
| *BNK* | 0.246278 | -0.702019 | -0.697641 | 1.000000 | | |
| *FDI* | 0.541272 | 0.458083 | 0.514252 | 0.087475 | 1.000000 | |
| *TRO* | 0.082638 | -0.478715 | -0.473564 | 0.539559 | 0.070177 | 1.000000 |

Source: Authors' own compilation.

## 4.2 Results of stationarity tests

Before embarking on the ARDL bounds testing, unit root properties of running variables are need to be checked. Several unit root tests are used by past studies to check data stationarity such as Augmented Dickey–Fuller (ADF) [73], Dickey-Fuller Generalized least squares (DF-GLS) [74], Kwiatkowski–Phillips–Schmidt–Shin [75] and Phillips-Perron (PP) [76]. The common drawback with these tests is that they do not provide adequate information about structural break points occurring in a series, hence, generate biased and spurious results. To pursue, we employ Zivot-Andrews test developed by Zivot and Andrews [77] in which they established three models to test the stationarity properties of time-series variables in a presence of structural break points. In addition to that, to scrutinize the level of integration of variables, we also employ the Ng-Perron test developed by Ng and Perron [78]. Unlike traditional unit root tests (e.g., PP, ADF), Ng-Perron outperforms the other tests and provides good results even when the sample data set is small [15]. In Ng–Perron tests, we reject the null hypothesis of non-stationarity if the critical value is greater than test statistics. This test construct four test statistics created upon detrended data $Y_t^d$ of generalized least square (GLS). These test statistics are the revised forms of Elliott, Rothenberg [74] point optimal statistics, Bhargava [79] $R_1$ statistics and Phillips and Perron [76] $Z_\alpha$ and $Z_t$ statistics.

Results of Ng-Perron unit root test are reported in Table 2 which shows that inflation and banking sector development are integrated at order of I(0) whereas market capitalization, GDP, FDI and trade openness are stationary at I(1). These results reveal that the variables selected for empirical investigation of the relationship between SMD and macroeconomic factors are integrated of mix order of integration. Like other traditional unit root tests, Ng-Perron test has a limitation that it does not account for structural break points in series which may lead towards biased and spurious results. To overcome this problem, we employ Zivot-Andrews unit root test and results are reported in Table 3 which indicates that market capitalization, inflation, and FDI are integrated at I(0) while rest are integrated at I(1). These results

**Table 2. Ng-Perron unit root test results.**

| | Stationarity of all variables at I(0) | | | | | Stationarity of all variables at I(1) | | | | |
|---|---|---|---|---|---|---|---|---|---|---|
| | Lag | $MZ_\alpha$ | $MZ_t$ | MSB | $MP_t$ | Lag | $MZ_\alpha$ | $MZ_t$ | MSB | $MP_t$ |
| *MC* | 0 | -10.82 | -2.28 | 0.21 | 8.66 | 1 | -28.92 | -3.80 | 0.13 | 3.15 |
| *GDP* | 0 | -6.02 | -1.72 | 0.29 | 15.11 | 0 | -18.38 | -2.91 | 0.16 | 5.66 |
| *INF* | 3 | -59.69 | -5.45 | 0.09 | 1.56 | 4 | -9.10 | -2.10 | 0.23 | 10.15 |
| *BNK* | 2 | -31.81 | -3.99 | 0.13 | 2.88 | 0 | -18.35 | -3.03 | 0.17 | 4.97 |
| *FDI* | 0 | -8.10 | -1.98 | 0.24 | 11.34 | 0 | -18.81 | -3.04 | 0.16 | 5.03 |
| *TRO* | 0 | -10.65 | -2.30 | 0.22 | 8.58 | 0 | -18.87 | -3.06 | 0.16 | 4.89 |
| **Critical Values** | **1%** | -23.80 | -3.42 | 0.143 | 4.03 | | | | | |
| | **5%** | -17.30 | -2.91 | 0.168 | 5.48 | | | | | |
| | **10%** | -14.20 | -2.62 | 0.185 | 6.67 | | | | | |

Note: Deterministic component is constant and trend. Lag selection is based on SC, whereas asymptotic critical values are taken from Table I of Ng and Perron [78].

are not consistent with Ng-Perron test; therefore, we rely on findings generated by Zivot-Andrews test and proceed further.

The differences in order of integration of variables provide an adequate reason to apply ARDL bounds testing approach as this technique can be applied even if all variables are not integrated of same order. To do so, the lag length of variables is first selected by using the Vector Autoregressive (VAR) lag order selection approach and results are presented in Table 4 as it is necessary to confirm the lag order at inception because ARDL *F*-statistics value is likely to be affected by lag length selection [15]. Based on results, we are indifferent in following any criteria as the optimal lag length suggested by all mentioned criteria is 1.

## 4.3 Main results of the ARDL bounds testing approach

The existence of long run relationship between SMD, economic growth, inflation, banking sector development, FDI and trade openness is examined by testing the joint significance for the null of no cointegration i.e., $H_0$: $\beta_{MC} = \beta_{GDP} = \beta_{INF} = \beta_{BNK} = \beta_{FDI} = \beta_{TRO} = 0$. Table 5 demonstrate the results of ARDL bounds testing procedure for cointegration while Table 6 exhibits the lower and upper bound critical values at different significance levels. Findings show that *F*-statistics value is 5.214, which is significantly higher than upper bound, *I(1)*, value reported by Pesaran, Shin [66], hence the null hypothesis of no-cointegration can be rejected. These findings validate the presence of long run relationship among running variables during the existence of structural break points in series during 1980 to 2019 in case of Pakistan.

**Table 3. Zivot-Andrews unit root test of variables in level and at the first difference.**

| | Stationarity of all variables in levels | | | | | | Stationarity of all variables at first differences | | | | | |
|---|---|---|---|---|---|---|---|---|---|---|---|---|
| | No Trend | Lag | Break Date | Trend | lag | Break Date | No Trend | Lag | Break Date | Trend | Lag | Break Date |
| *MC* | -4.228* | 0 | 1990 | -4.811* | 0 | 1988 | -8.856*** | 0 | 2008 | -8.598*** | 2 | 1999 |
| *GDP* | -2.901 | 1 | 2002 | -2.985 | 0 | 1986 | -6.430*** | 2 | 2018 | -6.858*** | 1 | 2008 |
| *INF* | -1.068 | 1 | 2003 | -6.099*** | 3 | 2014 | -4.569** | 0 | 2011 | -4.108* | 0 | 2009 |
| *BNK* | -4.074 | 0 | 2008 | -4.005 | 2 | 2008 | -5.376*** | 0 | 2004 | -5.114*** | 0 | 2012 |
| *FDI* | -3.185 | 0 | 1983 | -4.915** | 3 | 2007 | -6.240*** | 0 | 1985 | -6.120*** | 0 | 1986 |
| *TRO* | -3.213 | 0 | 1996 | -2.846 | 0 | 1992 | -7.724*** | 0 | 2000 | -6.845* | 0 | 2017 |

Note

*, ** and *** denote significance at 10%, 5% and 1% level.

**Table 4. Selection of optimal lag length.**

| VAR lag order selection criteria | | | | | | |
|---|---|---|---|---|---|---|
| Lag | LogL | LR | FPE | AIC | SC | HQ |
| **0** | 60.17845 | NA | 1.44e-10 | -2.798866 | -2.497205 | -2.691537 |
| **1** | 291.9069 | 365.8870* | 1.00e-14* | -12.41615* | -10.00287* | -11.55752* |
| **2** | 335.5610 | 52.84442 | 1.74e-14 | -12.13479 | -7.609880 | -10.52486 |

* indicates lag order selected by the criterion.

LR: sequential modified LR test statistic (each test at 5% level).

FPE: Final prediction error.

AIC: Akaike information criterion.

SC: Schwarz information criterion.

Having established that SMD and other explanatory variables are cointegrated with each other, we move towards the estimation of the model with ARDL bounds test approach. First, we determined the optimal lag length for the model by using the SC, i.e., ARDL (1, 0, 0, 1, 1, 0, 1). Long run and short run results of the selected model are presented in Table 7.

Empirical results shows that economic growth positively pronounce the SMD at 1% significance level in both long and short run. Long run coefficient of economic growth indicates that 1% change in GDP propels SMD by 2.546% if all else remains the same. This finding support the argument of Greenwood and Jovanovic [24], Greenwood and Smith [4], Boyd and Smith [64] and Garcia and Liu [80], that with the growth in Pakistan economy, the level of expenditure and saving increases, and ultimately more investors become inclined towards investing in capital market which propels stock market capitalization. Furthermore, these results are also in line with Atje and Jovanovic [81], Levine and Zervos [82], Levine and Zervos [60], Minier [83], Bayar [16], Shahbaz, Rehman [15], and Ho [48] who also confirm this positive relationship. In fact, stock market growth, in turn, further propels the economic growth. Past studies such as those carried out by Shahbaz, Ahmed [84] for Pakistan, Adjasi and Biekpe [20] and Agbloyor, Abor [85] for South Africa provide the evidence that SMD further enhances the economic growth in respective economies. These findings provide a direction to policy makers to formulate such economic policies that further enhance the development in both the economy and stock market. However, a lethargic increasing trend of only 6% is found in annual average economic growth from 2002 to 2018, and drastically it declines by 15% in 2019 [86]. Therefore, government should focus on economic growth along with establishing the investment confidence in economy in order to foster stock market capitalization.

On the inflation rate, results exhibit the significant inverse relationship between inflation rate and SMD in both long and short run. The long run results show that 1% increase in inflation rate inhibits the SMD by 1.675% if all else being identical. These findings are consistent with the conjecture that inflation and stock market capitalization are negatively associated with each other as also supported by other studies such as Boyd, Levine [87], Boyd, Levine

**Table 5. _F_-test results for ARDL bounds co-integration.**

| Dependent Variable | Function | _F_-Statistics | Cointegration Status |
|---|---|---|---|
| MC | F(MC \| GDP, INF, BNK, FDI, TRO) | 5.214** | Cointegrated |

Note

** denote significance at 5% level.

**Table 6. The critical values of the ARDL bounds test.**

| Level of significance | Lower Bound *I(0)* | Upper Bound *I(1)* |
|:---:|:---:|:---:|
| 1% | 1.99 | 2.94 |
| 5% | 2.27 | 3.28 |
| 10% | 1.99 | 2.94 |

Source: Pesaran et al. (2001) [66].

[35], Ben Naceur, Ghazouani [57] and Ho [13]. Furthermore, our results also in line with Akmal [88] and Shahbaz, Rehman [15] who also report that stocks in Pakistan work as shield against inflation. Based on these findings, policymakers should formulate strategies at national level to lower and stabilize inflation in order to bring investment flow towards capital markets which resultantly grow the economy. Besides that, empirical literature on economics have proved that independent and accountable state bank plays a significant role for financial stability and lower inflation in economy (see, [89,90]). In pursuant of this phenomenon, the State Bank Amendment Act 2021 has been recently passed by the National Assembly of Pakistan under which State Bank of Pakistan will work as sovereign authority and can take autonomous decisions for economic prosperity without any political influence of any government department [91]. The objectives of this amendment are domestic price stability and financial stability in the economy. Therefore, after implementation of this act it can be expected that inflation in Pakistan is likely to drop in upcoming periods and, in turn, the Pakistan stock market will develop.

The impact of banking and financial sector development on stock market capitalization in long run is found to be positive implying that financial development stimulates capital market development in Pakistan. It is found that a percent increase in banking sector development leads to 1.449% increase in stock market capitalization and vice versa, signifying towards the notion that financial sector development is a significant predictor of SMD. Although, the short-run results surprisingly exhibit the inverse relationship between SMD and banking sector development but the coefficient is insignificant. Several past studies confirm this complementary relationship between banking sector development and SMD ([13,18], see, [57,80,92]). Pakistan banking system has been transformed into efficient, strong and sound financial system after introducing the mega reforms initiated in early 1990s [93]. An assessment, carried out jointly by the international monetary fund (IMF) and The World Bank in 2004, conclude that improvements in the infrastructure of banking sector of Pakistan will substantially leads the economy towards growth and prosperity [94]. Therefore, the banking sector development, measured as domestic credit to private sector (% of GDP), is noticeably improved from 2002 to 2008 by 31.7%. However, a persistent radical downfall of 8.7% is found from 2008 to 2015 which is most likely due to global financial crises [86]. Nevertheless, our results dictate the policy makers to formulate policies that promote financial sector development which, in turn, grow the stock market.

Results show that SMD and FDI are negatively associated with each other in long run and positively in short-run. The long run findings confirm the theoretical economic view that in developing economies like Pakistan, FDI works as a substitute for SMD. Long run results show that a percent increase in FDI hinder the SMD by 0.665%. In Pakistan, persistent rise in FDI is found from 2004 to 2007 with the average increasing rate of 36%. After that, it surprisingly continues to drop till 2012 with the average decreasing rate of 62%. Particularly, the bothersome situation is that 45% serious downfall in FDI is appeared during the fiscal year of 2008–2009 and 2009–2010 [7]. Although FDI indirectly foster capital market through economic

growth-enhancing effect [52], but this negligible effect of FDI is inadequate to accelerate the performance of SMD in Pakistan which is possible due to bad economic conditions, terrorism and adverse governance.

Finally, on the trade openness, our results are surprisingly contrary to the conventional wisdom, that the effect of trade openness on the SMD is found to be negative and significant at α = 5%. However, these findings are in line with the studies conducted by those such as Ho [13] and Shahbaz, Rehman [15]. It is found that a 1% increase in trade openness hamper the SMD by 0.747%. The inverse relationship between two can be explained by the level of trade in Pakistan. As argued by Do and Levchenko [95], the financial development (including stock market development) in an economy is affected by the comparative advantage in trade. They demonstrate that if main exports of the economy significantly depend on internal finance then, in result, the pattern of financial development growth becomes slow down. In Pakistan, the top three export during 2018–2019 were worn clothing, knit clothing and cotton which accounted for 18.3%, 14.1% and 13.5% of total exports respectively [96]. In general, top ten exports of Pakistan accounted for 73.5% of the overall value of its global shipment during the same period [97]. Particularly, top exporting companies of Pakistan are involved in cross-border trade are concentrate on products related to textiles, leather, sports, and chemicals [98]. These facts assert that heavy reliance of major exports in internal finance may justify the inverse association between trade openness and stock market development.

Next, the short run dynamic effect of macroeconomic variables on SMD is examined and their findings are reported in Table 7. The coefficient of $ECT_{t-1}$ is negative and statistically significant. It tells that if the variables are drift away in short-run from the level of equilibrium by 1%, they will move back towards adjustment by 74.3% per year. Furthermore, high $R^2$ of approximately 84% indicates that the selected ARDL model fits well.

## 4.4 Diagnostic and stability tests

In order to assess the validity of our model, we perform some diagnostic tests and their results are illustrated in Table 8. As informed by our results, data in the model is normally distributed,

**Table 7. The long run and short-run results of the selected model.**

| Regressor | Coefficient | Standard Error | t-statistics | p-value |
|---|---|---|---|---|
| *Long run results (dependent variables is MC)* | | | | |
| GDP | 2.546*** | 0.229 | 3.848 | 0.001 |
| INF | − 1.675*** | 0.166 | 12.099 | 0.000 |
| BNK | 1.449** | 0.014 | −2.235 | 0.033 |
| FDI | − 0.665*** | 0.013 | -4.548 | 0.000 |
| TRO | − 0.747** | 0.011 | 2.338 | 0.026 |
| *Short-run results (dependent variables is ΔMC)* | | | | |
| ΔGDP | 2.192*** | 0.101 | 3.114 | 0.004 |
| ΔINF | −0.872*** | 0.123 | 8.980 | 0.000 |
| ΔBNK | 1.349 | 0.101 | 1.253 | 0.219 |
| ΔFDI | 0.726*** | 0.101 | 3.080 | 0.004 |
| ΔTRO | −2.837 | 1.028 | −0.626 | 0.536 |
| ECT (-1) | −0.743*** | 0.129 | 2.976 | 0.006 |

**Notes**: R-squared = 0.8413, Adjusted R-squared = 0.8257, S.E. of regression = 0.2999, AIC = 0.628451, SC = 0.12350, F-statistic = 6.707061, Prob(F-statistic) = 0.0000.

Notes

*,**,*** denotes significance level at the 10%, 5% and 1% respectively. Δ denotes first difference operator.

**Table 8. Results of robustness tests.**

| Test | Statistic | *p*-value |
|---|---|---|
| Jarque-Bera Normality Test | 2.044 | 0.4485 |
| Ramsey RESET Test: F (1, 33) | 0.709 | 0.4058 |
| Autocorrelation | 1.181 | 0.3200 |
| Heteroscedasticity | 1.678 | 0.1663 |

and free from autocorrelation problem. In addition to that, we do not find heteroskedasticity and function misspecification problem in our model. Furthermore, the stability of the long and short run coefficients is examined through cumulative sum of recursive residuals (CUSM) and cumulative sum of squares of recursive residuals (CUSMSQ) tests and their graphs are plotted in Figs 1 and 2 respectively. Both figures confirm the stability test for our model estimates and proved that model is correctly specified.

## 5. Conclusion and policy implications

The capital market in an economy is deemed as a significant driver of economic growth and prosperity. More investment and business opportunities can be introduced in economy through an established and well-organized stock market by mobilizing the savings and mitigating business risk. Over the past two decades, the emerging capital markets have been globalized and deepened with the time, and consequently, the expansion in terms of capitalization

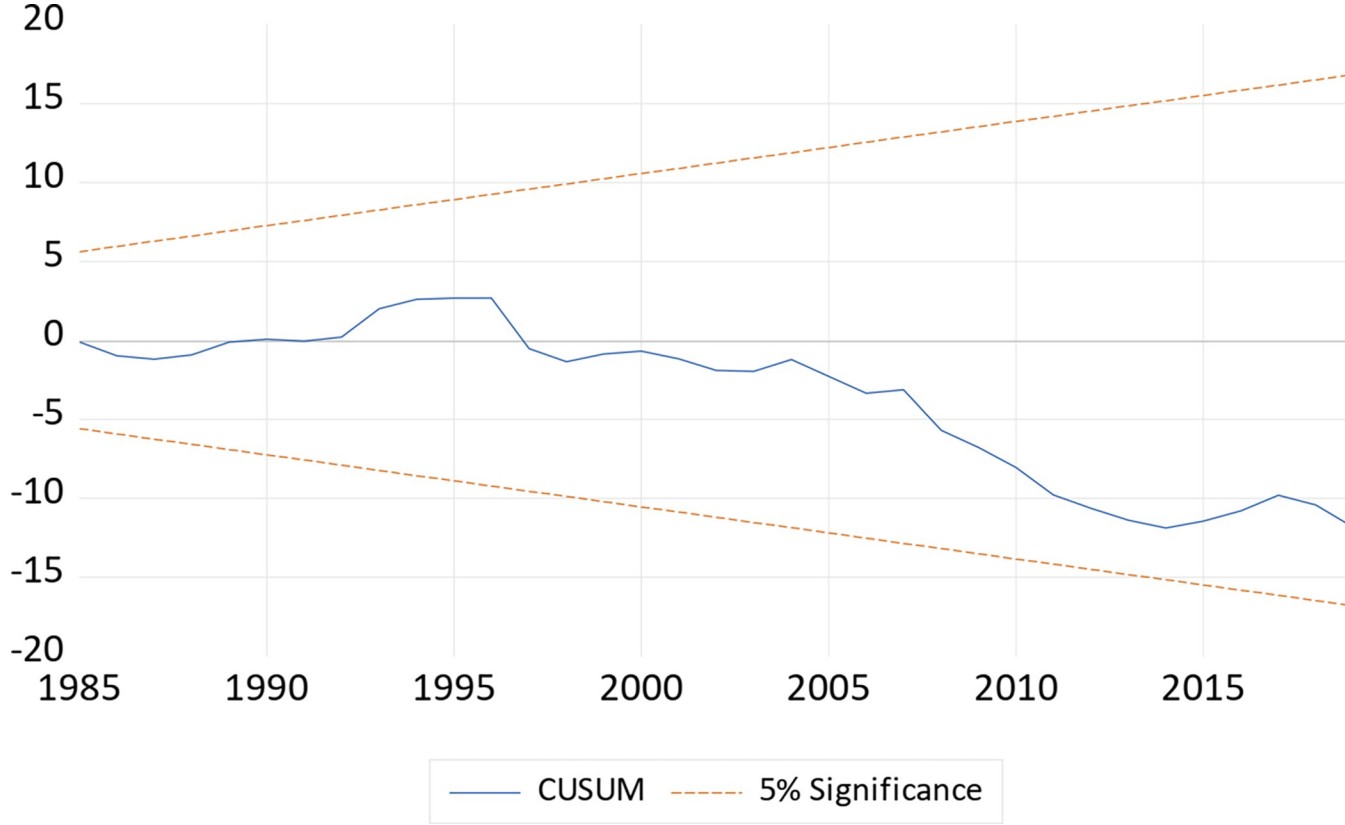

**Fig 1. Plot of cumulative sum of recursive residuals.**

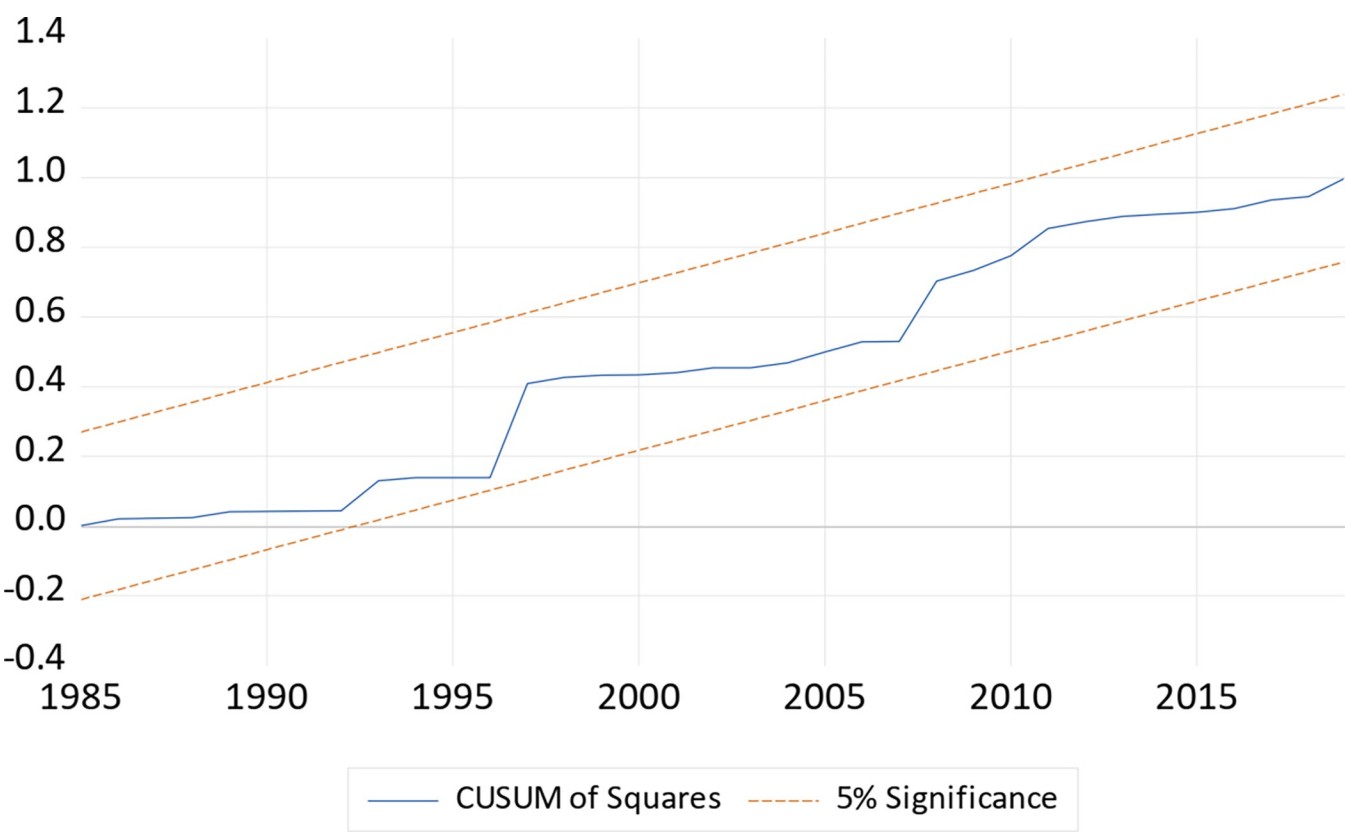

**Fig 2. Plot of cumulative sum of squares of recursive residuals.**

and landscape of these markets exhibit substantial change in a set of investing opportunities. In addition to that, a well-established stock market contributes in an economy by improving corporate governance practices, assisting in global and domestic risk diversification, and fostering the assets' liquidity in financial market.

Given the importance of stock market, this study investigates the influence of major macroeconomic variables on SMD in case of Pakistan in presence of unknown structural breaks over the period of 1980–2019 using annual time-series data. While exploring the determinants of SMD, we used ARDL bounds testing cointegration approach to examine the long and short run relationship among variables along with the Ng-Perron and Zivot-Andrews test to confirm the presence of unit root and the order of integration of all variables. Results confirm the cointegration among variables. It is found that GDP is directly while inflation is inversely and significantly linked with SMD in both long and short run. Banking sector development positively and significantly affect SMD in long run but no effect in short run. FDI is negatively associated with SMD in long run but surprisingly positively associated in short run. Trade openness negatively pronounce the SMD in long run but no impact in short run. Finally, the negative and significant coefficient of ECT signifies towards the notion that if, in short run, the series drift away by 1% from the level of equilibrium, they will adjust back by 74.3% in a year. The findings of this study have some policy implications. Policy makers, for example, should develop such economic and financial policies that further foster the economic growth which, in turn, promote the stock market. Similarly, government of Pakistan should control the inflation rate which in turn put a favorable effect on stock market capitalization. Law and enforcement authorities should account for terrorism and adverse governance system and ensure political

stability in country in order to gain foreign investors' trust and to bring foreign capital flows into the stock market.

## Acknowledgments

We thank to the two anonymous reviewers for their careful reading of our manuscript and their many insightful comments and valuable suggestions.

## Author Contributions

**Conceptualization:** Abdullah Bin Omar.

**Data curation:** Abdullah Bin Omar, Anis Ali.

**Formal analysis:** Robina Kouser.

**Methodology:** Anis Ali, Mamdouh Abdulaziz Saleh Al-Faryan.

**Project administration:** Salma Mouneer, Robina Kouser.

**Resources:** Mamdouh Abdulaziz Saleh Al-Faryan.

**Software:** Anis Ali, Salma Mouneer.

**Supervision:** Abdullah Bin Omar, Salma Mouneer, Robina Kouser.

**Validation:** Salma Mouneer, Mamdouh Abdulaziz Saleh Al-Faryan.

**Visualization:** Anis Ali, Robina Kouser.

**Writing – original draft:** Abdullah Bin Omar, Robina Kouser.

**Writing – review & editing:** Abdullah Bin Omar.

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
