## [Decision Letter · Decision Letter 0]

21 Jul 2022

PONE-D-22-14460Is stock market development sensitive to macroeconomic indicators? A fresh evidence using ARDL bounds testing approachPLOS ONE

Dear Dr. Omar,

Thank you for submitting your manuscript to PLOS ONE. After careful consideration, we feel that it has merit but does not fully meet PLOS ONE’s publication criteria as it currently stands. Therefore, we invite you to submit a revised version of the manuscript that addresses the points raised during the review process.

We look forward to receiving your revised manuscript.

Kind regards,

Aurelio F. Bariviera, Ph.D.

Academic Editor

PLOS ONE

Journal Requirements:

2. Please ensure that you include a title page within your main document. You should list all authors and all affiliations as per our author instructions and clearly indicate the corresponding author.

Reviewers' comments:

Reviewer's Responses to Questions

**Comments to the Author**

1. Is the manuscript technically sound, and do the data support the conclusions?

Reviewer #1: Yes

Reviewer #2: Partly

2. Has the statistical analysis been performed appropriately and rigorously? 

Reviewer #1: Yes

Reviewer #2: Yes

3. Have the authors made all data underlying the findings in their manuscript fully available?

Reviewer #1: Yes

Reviewer #2: Yes

4. Is the manuscript presented in an intelligible fashion and written in standard English?

Reviewer #1: Yes

Reviewer #2: Yes

5. Review Comments to the Author

Reviewer #1: The financial market is the key of the sources to improve the prosperty. Totally they will testify this relationship in their manuscript. My decision is to accept the paper for publication to show more evidences to improve the finance and the economy connections.

Reviewer #2: Dear author(s),

Please find below my comments that you may find them useful.

Introduction:

Given there are studies examining macroeconomic factors of stock market development in Pakistan, particularly Shahbaz et al. (2016) with similar model specification, estimation technique and empirical findings, I cannot see any significant contribution in this study. Therefore, you should revise your study to provide a strong motivation on how your study is contribution to the existing body of knowledge.

Since your discussion on Pakistan include the development of COVID-19 in 2020 2021, I suggest you extend your analysis to 2020 or 2021 if data allow. Currently, your study ended in 2019, the discussion on stock market development in 2020 and 2021 became irrelevant. On the discussion of COVID-19, I would expect you research to capture the influence of COVID-19 on stock market development in your model.

Literature review:

On the literature review, you should have a brief theoretical review on how macroeconomic determinants influence stock market development. See Ho and Iyke (2017).

Line 124, Ho is she.

Methodology:

Equations 1 and 2 are wrongly specified. Apart for the dependent variable (MC) being lagged as the independent variables, all the other independent variables are not lagged. This is the idea of ARDL.

You should motivate why there is dummy variable in the equation. What is it for? Why it is required given the economic environment in Pakistan during your study period.

On economic growth, neither the source your cited nor any existing theories suggest that the relationship is negative (substitutes). Please verify.

Results and Discussion:

In addition to the main results, you should conduct robustness check in your analysis.

Since the negative relationship between trade openness and stock market development is against our conventional wisdom, you should clearly explain why trade openness negatively affects stock market development with possible theoretical explanation in the context of Pakistan. Also, the concept of trade and foreign portfolio investment should not be mixed in your discussion.

Reference:

Ho, S. Y., & Iyke, B. N. (2017). Determinants of stock market development: a review of the literature. Studies in Economics and Finance.

Shahbaz, M., Rehman, I. U., & Afza, T. (2016). Macroeconomic determinants of stock market capitalization in an emerging market: fresh evidence from cointegration with unknown structural breaks. Macroeconomics and Finance in Emerging Market Economies, 9(1), 75-99.

6. PLOS authors have the option to publish the peer review history of their article (what does this mean?). If published, this will include your full peer review and any attached files.

Reviewer #1: No

Reviewer #2: No

---

## [Author Response · Author response to Decision Letter 0]

14 Aug 2022

Hi 

Dear Reviewers, 

I have incorporated all mentioned changes/corrections and addressed all points which you raised. Please have a look on updated files.

Regards

---

## [Decision Letter · Decision Letter 1]

4 Sep 2022

PONE-D-22-14460R1Is stock market development sensitive to macroeconomic indicators? A fresh evidence using ARDL bounds testing approachPLOS ONE

Dear Dr. Omar,

Thank you for submitting your manuscript to PLOS ONE. After careful consideration, we feel that it has merit but does not fully meet PLOS ONE’s publication criteria as it currently stands. Therefore, we invite you to submit a revised version of the manuscript that addresses the points raised during the review process.

We look forward to receiving your revised manuscript.

Kind regards,

Aurelio F. Bariviera, Ph.D.

Academic Editor

PLOS ONE

Journal Requirements:

Additional Editor Comments :

Please revise equations 1 and 2 as required by Reviewer 2.

Reviewers' comments:

Reviewer's Responses to Questions

**Comments to the Author**

1. If the authors have adequately addressed your comments raised in a previous round of review and you feel that this manuscript is now acceptable for publication, you may indicate that here to bypass the “Comments to the Author” section, enter your conflict of interest statement in the “Confidential to Editor” section, and submit your "Accept" recommendation.

Reviewer #1: All comments have been addressed

Reviewer #2: All comments have been addressed

2. Is the manuscript technically sound, and do the data support the conclusions?

Reviewer #1: Yes

Reviewer #2: Yes

3. Has the statistical analysis been performed appropriately and rigorously? 

Reviewer #1: Yes

Reviewer #2: Yes

4. Have the authors made all data underlying the findings in their manuscript fully available?

Reviewer #1: Yes

Reviewer #2: Yes

5. Is the manuscript presented in an intelligible fashion and written in standard English?

Reviewer #1: Yes

Reviewer #2: Yes

6. Review Comments to the Author

Reviewer #1: Accept as it is. After the revision the manuscript add the valuable contributions on the subject of positive impact and the connection with the economic growth and banking sector.

Reviewer #2: Dear author(s),

Most of the comments have been addressed satisfactorily except the following one.

5. Methodology: Equations 1 and 2 are wrongly specified. Apart for the dependent variable

(MC) being lagged as the independent variables, all the other independent variables are not

lagged. This is the idea of ARDL.

Compliance:

The basic and standard equation of ARDL bounds testing is given below (as a screenshot). As you can see that dependent variable (in short- and long-run terms) is being lagged accordingly. Therefore, the equations 1 and 2 given in manuscript are correct. Furthermore, you can explore Ho (2019, pp. 328-329) and (Shahbaz et al., 2016, p. 83) to confirm that our ARDL econometric form is correctly specified.

Reviewer’s response:

In your model specification, apart from the dependent variable (MC) that was lagged as independent variable (as shown by the summation from 1 to k), all other independent variables should not be lagged (as shown by the summation from 0 to k). Please go and revise your equations 1 and 2.

7. PLOS authors have the option to publish the peer review history of their article (what does this mean?). If published, this will include your full peer review and any attached files.

Reviewer #1: No

Reviewer #2: No

---

## [Author Response · Author response to Decision Letter 1]

20 Sep 2022

Hi, 

Dear Reviewers, 

I have corrected the equations as prescribed in email/decision letter. 

Regards

---

## [Editor Report · Decision Letter 2]

21 Sep 2022

Is stock market development sensitive to macroeconomic indicators? A fresh evidence using ARDL bounds testing approach

PONE-D-22-14460R2

Dear Dr. Omar,

We’re pleased to inform you that your manuscript has been judged scientifically suitable for publication and will be formally accepted for publication once it meets all outstanding technical requirements.

Kind regards,

Aurelio F. Bariviera, Ph.D.

Academic Editor

PLOS ONE

Additional Editor Comments (optional):

The authors made the corrections prescribed by Reviewer 2 in the Revision 1
---

## [Editor Report · Acceptance letter]

4 Oct 2022

PONE-D-22-14460R2 

Is stock market development sensitive to macroeconomic indicators? A fresh evidence using ARDL bounds testing approach 

Dear Dr. Omar:

I'm pleased to inform you that your manuscript has been deemed suitable for publication in PLOS ONE. Congratulations! Your manuscript is now with our production department. 

Kind regards, 

on behalf of

Dr. Aurelio F. Bariviera 

Academic Editor

PLOS ONE